# An Experimental Approach to Address the Functional Relationship between Antioxidant Enzymes and Mitochondrial Respiratory Complexes

**DOI:** 10.3390/mps6020032

**Published:** 2023-03-24

**Authors:** Daniela Mendes, Ana Maria Silva, Maria Manuel Oliveira, Paula B. Andrade, Romeu A. Videira

**Affiliations:** 1REQUIMTE/LAQV, Laboratory of Pharmacognosy, Department of Chemistry, Faculty of Pharmacy, University of Porto, Rua de Jorge Viterbo Ferreira, nº 228, 4050-313 Porto, Portugal; mendesdaniela04@hotmail.com (D.M.); pandrade@ff.up.pt (P.B.A.); 2Department of Life Sciences, University of Coimbra, Calçada Martim de Freitas, 3000-456 Coimbra, Portugal; anamariapsilva@gmail.com; 3Chemistry Center—Vila Real (CQ-VR), Chemistry Department, School of Life and Environmental Sciences, University of Trás-os-Montes e Alto Douro, UTAD, 5001-801 Vila Real, Portugal; mmso@utad.pt

**Keywords:** mitochondria, respiratory complex activity assays, cytosolic redox state, antioxidant enzymes activity assays

## Abstract

Mitochondrial dysfunction and cytosolic oxidative stress are pathological biomarkers interlinked in several chronic diseases and cellular toxicity promoted by high-energy radiation or xenobiotics. Thus, assessing the activities of the mitochondrial redox chain complexes and the cytosolic antioxidant enzymes in the same cell culture system is a valuable approach to addressing the challenge of chronic diseases or unveiling the molecular mechanisms underlying the toxicity of physical and chemical stress agents. The present article gathers the experimental procedures to obtain, from isolated cells, a mitochondria-free cytosolic fraction and a mitochondria-rich fraction. Furthermore, we describe the methodologies to evaluate the activity of the main antioxidant enzymes in the mitochondria-free cytosolic fraction (superoxide dismutase, catalase, glutathione reductase and glutathione peroxidase), and the activity of the individual mitochondrial complexes I, II and IV, as well as the conjugated activity of complexes I–III and complexes II–III in the mitochondria-rich fraction. The protocol to test the citrate synthase activity was also considered and used to normalize complexes. The procedures were optimized within an experimental setup to allow that each condition to be tested only requires sampling of one T-25 flask of cells 2D cultured, as the typical results presented and discussed here.

## 1. Introduction

Mitochondrial dysfunction and oxidative stress are typical pathological hallmarks of many diseases and cellular toxicity biomarkers triggered by different external stimuli, including high-energy radiation, environmental pollutants and pharmacological drugs [1,2,3]. Mitochondrial functionality is coupled with the cell redox state since all mitochondrial respiratory chain complexes can transfer electrons to molecular oxygen (O_2_). Unlike Complex IV, which reduces O_2_ to water, the other complexes (mainly Complexes I and III) release reactive oxygen species (ROS; superoxide anion radical (O_2_^•−^)/hydrogen peroxide (H_2_O_2_)) with an impact on cellular redox status (Figure 1) [4,5,6]. Despite these mitochondria-dependent ROS having a pivotal role in cell signaling cascades, often, mitochondrial dysfunction leads to ROS generation at levels that overpass not only the required concentrations for cell signaling under physiological conditions but also the capacity of the antioxidant defense system to suppress them, thus promoting oxidative stress. On the other hand, several mitochondrial components, including mitochondrial DNA and inner-membrane phospholipids, are especially vulnerable to oxidative damage. The accumulation of oxidized lipids and damaged DNA worsens mitochondrial dysfunction, amplifying oxidative stress’s harmful cellular effects [7,8,9]. Thus, mitochondrial dysfunction and oxidative stress can work under a vicious cycle that compromises organisms’ cell functions and physiological performance.

Developing methodologies to establish the relationship between mitochondrial functionality and cytosolic cellular redox state is a common challenge for researchers in many biomedical sciences fields. The present work outlines a strategy to assess the mitochondrial redox chain complexes and cytosolic antioxidant enzyme activities in a cell culture system. The strategy involves a cryopreservation step followed by subcellular fractionation by differential centrifugation to obtain mitochondria-free cytosolic and mitochondria-rich fractions. Mitochondria-free cytosolic fraction is used to assess the activity of some antioxidant enzymes, while the mitochondria-rich fraction is used to evaluate the activity of mitochondrial respiratory chain complexes. The experimental procedures to assess the activity of the enzymes are detailed together with theoretical background, chemical reactions and a colorful ribbon to illustrate the detected colorful changes.

## 2. Procedure and Experimental Design

### 2.1. Cell Fractionation: Mitochondria-Free Cytosolic Fraction and Mitochondria-Rich Fraction

Studies with cell cultures are the first in vitro choice to address the metabolic pathways and cell signaling cascades under physiological and pathological conditions, particularly relevant in the toxicological and pharmacological context. The experimental procedures described below were optimized with neuroblastoma SH-SY5Y (ATCC) cells, and they were also tested with human fibroblast BJ (ATCC), 3T3-L1 preadipocytes (ATCC, CL-173) and murine microglial BV-2 (ATCC, E0C2, CRL-2467) cells. They are robust enough to suit for most cell types growing in culture. Thus, the methodology is presented to assess mitochondrial redox chain complexes activity and the cytosolic antioxidant enzymes in SH-SY5Y cells.

Neuroblastoma SH-SY5Y cells were cultured in Dulbecco’s Modified Eagle Medium/Nutrient Mixture F-12 (DMEM/F-12) supplemented with Fetal Bovine Serum (FBS, 10%) and Penicillin-Streptomycin (Pen-Strep, 1%). Cells were maintained in a carbon dioxide incubator (Toreuse model 2428; Saint Louis, MO, USA), using the standard conditions with a controlled humidified atmosphere, carbon dioxide and temperature. For assays, SH-SY5Y cells were seeded in T-25 flasks (30,000/cm^2^) to obtain, 48 h after incubation, a cell number suitable for experiments (about 1.5 × 10^6^ cells/flask). Alternatively, 24 h after incubation, cells were treated and re-incubated for 24 h to obtain, under control conditions, cultures with the cell number mentioned above. To stop cell metabolism, the cell culture medium was removed, washed with HBSS and the cells in the flask are frozen in liquid nitrogen and stored at −80 °C until to be used.

The frozen adherent cells were collected from the flasks by scraping after adding 0.5 mL of homogenization buffer (Sucrose 130 mM, KCl 50 mM, MgCl_2_ 5 mM, KH_2_PO_4_ 5 mM, HEPES 5 mM, pH 7.4) at 4 °C, and homogenized by using a 3 mL Glass-Teflon Potter Elvehjem homogenizer (Merck, Reference P7734-1EA; clearance of 0.004–0.006 in.) using 20 up- and downstrokes at 500 rpm. The mitochondria-free cytosolic and mitochondria-rich fractions were obtained by sequential differential centrifugation steps, as displayed in Figure 2, using standard procedures [10]. Typically, about 1 mL of mitochondria-free cytosolic fraction with a protein concentration of 1.0 mg/mL was obtained. In comparison, the mitochondria-rich fraction was collected in 200 µL with a protein concentration of about 0.5 mg/mL. The efficacy of Glass-Teflon Potter Elvehjem homogenizer to promote cell lysis was ensured measuring the lactate dehydrogenase (LDH) activity in cell homogenate, cell pellet reconstituted in 1 mL of buffer solution supplemented with 2% of Triton X-100, and in the supernatant fraction [11]. The procedure was suitable since LDH activity was not detected in 40 µL of the reconstituted pellet and more than 95% of this enzyme activity exhibited by cell homogenate is preserved in the supernatant fraction. As indicated in Figure 2, citrate synthase activity was assessed in supernatant, and mitochondria-free cytosolic fraction in addition to mitochondria-rich fraction to evaluate the purity of the fractionation methodology. In fact, the activity of this mitochondrial enzyme was not detected in the mitochondria-free cytosolic fraction and the activity detected in supernatant was preserved in the mitochondria-rich fraction.

### 2.2. Mitochondria-Free Cytosolic Fraction: Assessment of the Activity of the Cytosolic Antioxidant Enzymes

The activities of cytosolic antioxidant enzymes: superoxide dismutase (Cu/Zn-SOD), glutathione reductase (GR) and glutathione peroxidase (GPx) were evaluated in 250 µL of reaction medium by using a microplate reader (allowing several assays at the same time (Synergy H1 Biotek, Winooski, VT, USA)) while catalase (CAT) activity was assessed polarographically following the oxygen production in 1 mL reaction medium by using a Clark-type electrode (Hansatech, Norfolk, UK). All assays were conducted at 37 °C physiological temperature for humans, using a volume of the mitochondria-free cytosolic fraction containing 50 µg of protein. For each enzyme, assay conditions are summarized in a table, including the volumes used for each reagent/biological sample (Tables 1–4). The compounds used in the assessment of the cytosolic antioxidant enzymes had an analytic degree (purity ≥ 99%) and were purchased in SIGMA/MERCK group.

#### 2.2.1. Superoxide Dismutase (Cu/Zn) Activity

Suspension of mitochondria-free cytosolic fraction (1000 µg protein/mL);Phosphate buffer (KH_2_PO_4_ 100 mM, EDTA 5 mM, pH 7.4);Nitroblue tetrazolium chloride (NBT, 2 mM in phosphate buffer);Xanthine oxidase (12.5 U/mL in phosphate buffer);Xanthine (2 mM in phosphate buffer).

Cu/Zn-SOD activity was assessed indirectly based on competition for O_2_^•−^, generated by the xanthine/xanthine oxidase system, between the enzyme and the yellow NBT that reacted with the O_2_^•−^ to produce purple formazan, detectable by changes in absorption at 560 nm, as illustrated in Figure 3 [10,12]. Thus, the time-dependent absorbance increased, at 560 nm in the absence and presence of mitochondria-free cytosolic fraction was recorded in a microplate reader according to the conditions in Table 1. Cu/Zn-SOD activity is expressed as a unit of the enzyme’s catalytic activity (U) per milligram protein. When the O_2_^•−^ was generated by 1 U of xanthine oxidase, one catalytic unit of Cu/Zn-SOD is defined by the amount of biological sample required to decrease the slope of NBT reduction by 50%. Since 0.5 U of xanthine oxidase was used in the present assay conditions, the biological sample required to decrease the slope of NBT reduction by 50% corresponds to 0.5 U of Cu/Zn-SOD activity.

#### 2.2.2. Glutathione Reductase Activity

Suspension of mitochondria-free cytosolic fraction (1000 µg protein/mL);Phosphate buffer (KH_2_PO_4_ 100 mM, EDTA 5 mM, pH 7.4);GSSG (10 mM in water);NADPH (10 mM in water).

GR activity was evaluated spectrofluorimetrically following NADPH oxidation associated with the reduction of GSSG to GSH. As illustrated in Figure 4, this was detected by the blue/cyan fluorescence decay. The time-dependent decrease in NADPH fluorescence intensity at 450 nm (setting excitation at 366 nm) promoted by the mitochondria-free cytosolic fraction in the absence and presence of GSSG (enzyme-substrate) was recorded in a microplate reader according to the conditions displayed in Table 2. GR activity was determined by the difference between the slopes of NADPH oxidation in the presence and absence of GSSG and expressed in nmol NADPH per minute per milligram of protein. To convert the slopes of NADPH fluorescence decay promoted by GR activity in moles, a calibration curve of the NADPH fluorescence intensity as a function of concentration prepared in the same assay conditions is required. It is important to highlight that NADPH oxidation can also be assessed by the absorbance decrease at 340 nm (ε_NADPH_ = 6220 M^−1^ cm^−1^) [10,13]. However, the fluorescence-based methodology was selected since, under the above-mentioned conditions, NADPH is a strong fluorophore while its oxidized pair (NADP^+^) is non-fluorescent, and for analysis of biological samples, in general, the fluorescence-based quantification is more sensitive than UV-based absorption measurement [14].

#### 2.2.3. Glutathione Peroxidase Activity

Suspension of mitochondria-free cytosolic fraction (1000 µg protein/mL);Phosphate buffer (KH_2_PO_4_ 100mM, EDTA 5 mM, pH 7.4);GSH (10 mM in water);GR (116.48 U/mL in phosphate buffer);NADPH (10 mM in water);H_2_O_2_ (88 mM in water).

GPx activity in the mitochondria-free cytosolic fraction was evaluated through a coupled reaction with the GR enzyme. GPx reduced H_2_O_2_ to water in this assay and oxidized GSH to GSSG [10,15]. The generated GSSG was reduced to GSH with NADPH consumption by GR, which can be followed by fluorescence decay as described above (Figure 5). Under GR excess conditions, the NADPH oxidation is limited by GPx activity and can be easily measured spectrofluorimetrically. Thus, the time-dependent decrease in NADPH fluorescence intensity at 450 nm (setting excitation at 366 nm) promoted by the mitochondria-free cytosolic fraction in the absence and presence of GSH (substrate of GPx enzyme) was recorded in a microplate reader, as displayed in Table 3. GPx activity was determined, in the first step, by the difference between the slopes of NADPH oxidation in the presence and absence of GSH, expressed in nmol NADPH per minute per milligram of protein. A calibration curve of the NADPH fluorescence intensity as a function of concentration was prepared in the same assay conditions to convert the slopes of NADPH fluorescence decay promoted by the GR/GPx coupled system in mol. Considering the stoichiometry of the reactions, the amount of NADPH used by GR is equal to the amount of H_2_O_2_ used by GPx. Thus, the enzyme activity can be expressed in nmol H_2_O_2_ per minute per milligram of protein in the final step.

#### 2.2.4. Catalase Activity

Suspension of mitochondria-free cytosolic fraction (1000 µg protein/mL);Phosphate buffer (KH_2_PO_4_ 100 mM, EDTA 5 mM, pH 7.4);H_2_O_2_ (88 mM in water).

CAT reduces H_2_O_2_ to water and O_2_ (colorless reaction, Figure 6). CAT activity was determined polarographically following O_2_ production for 2–3 min, using a Clark-type oxygen electrode, resulting from H_2_O_2_ decomposition (Table 4) [16,17]. The difference between the slopes in the presence and absence of mitochondria-free cytosolic fraction determines CAT activity. It is expressed in terms of nmol O_2_ produced per minute per milligram of protein. CAT activity can be also assessed by the absorbance decrease at 240 nm [18]. We propose the assessment of CAT activity polarographically, since it is considered an accurate method, and compatible with complex biological samples, avoiding the matrix-related drawbacks underlying the UV-based absorption procedure [16]. Additionally, Hansatech Oxygraph Instrument allows to detect at each second changes in oxygen concentration of 0.01 nmol/mL range. Under our assay conditions, the 10-fold decrease in the biological sample (i.e., to 5 mg protein/mL) still allows reliably detection of CAT activity, as shown in the Appendix A (O_2_ production rate = 22.92 nmol/mL/min., corresponding to H_2_O_2_ consumption rate of 45.84 nmol/mL/min according with stoichiometric of the CAT-catalyzed reaction).

### 2.3. Mitochondria-Rich Fraction: Assessment of the Mitochondrial Respiratory Chain Complexes Activities

The mitochondrial electron transport chain is supplied by NADH at the level of Complex I and by succinate at the level of Complex II (a citric acid cycle enzyme linked to the inner membrane). The membrane pool of ubiquinone accepts the electrons from Complex I or Complex II. It delivers them to Complex III, reducing cytochrome *c* (Cyt *c*), a small protein that transports electrons for Complex IV, where the O_2_ is reduced to water. It is reported for Mammalia mitochondria that the transmembrane complexes I, III and IV can occurs as individual identities as well as organized in super-complexes with well-defined proportions. Complex II, mainly committed to the citric acid cycle that occurs in the matrix, occurs preferentially as individual enzyme exhibiting low tendency to integrate the super-complexes [19,20]. Thus, to obtain an overview of the activity of the respiratory chain enzymes, we propose assessing the activity of individual Complexes I, II and IV, which can be activated by the external addition of their endogenous water-soluble substrates. The assessment of complex III activity represents a different challenge since its substrate (ubiquinone) is not soluble in water. It can also occur associated with Complex I (Super-complex I–III) within a process that recruits its ubiquinone pool. Thus, we propose assessing the Super-complex I–III activity by using NADH for activation and measuring the time-dependent Cyt *c* reduction. To obtain an overview of the activity of Complex III not assembled into the Super-complexes, we propose to assess the Complex II–III conjugated activity, using succinate to activate Complex II and follow the reduction of Cyt *c*. The mitochondria-rich fraction was submitted to three cycles of freezing/thawing, before assaying mitochondrial enzyme activities, to allow enzyme substrates to reach the active site of membrane redox chain complexes located on the mitochondrial matrix side such as Complexes I and II. While dodecyl maltoside can be also used for this propose, the procedure requires a progressive titration to find the minor amount of detergent necessary to permeabilize the inner membrane without disrupting its bilayer organization, which has physiological relevance [21]. Considering the amount of biological sample available, this type of procedure may compromise the feasibility of assessing the activity of the mitochondrial respiratory complexes. The activity of the mitochondrial redox chain complexes is normalized by citrate synthase (CS) activity that is tightly correlated with morphometric data [22]. This strategy eliminates the putative differences in the mitochondria richness of the several preparations used to assess the enzyme activities [17]. The activity of the mitochondrial enzymes was assessed by spectrophotometric or spectrofluorimetric assays, at 37 °C, in 250 µL of reaction medium supplemented with a volume of the mitochondria-rich fraction containing 10µg of protein. Using a microplate reader (Synergy H1 Biotek), we had the advantage of the assessment of several assays at the same time. For each mitochondrial enzyme complex and CS, a table with assay conditions, including the volumes used for each reagent/biological sample, is presented (Tables 5–11). All the compounds used in the assessment of the mitochondrial respiratory chain complexes have analytic degree (purity ≥ 99%) and were purchased in SIGMA/MERCK group.

#### 2.3.1. Complex I: NADH:Ubiquinone Oxidoreductase Activity

Suspension of mitochondria-rich fraction (0.5 mg protein/mL);Phosphate buffer (KH_2_PO_4_ 25 mM, MgCl_2_ 5 mM, pH 7.5);Potassium cyanide (KCN, 100 mM in water);Antimycin A (2 mM in ethanol);Decylubiquinone (38.75 mM in DMSO);Rotenone (600 µM in ethanol);NADH (5 mM in water).

Complex I was evaluated spectrofluorimetrically following NADH oxidation, detected by the blue/cyan fluorescence decay, as illustrated in Figure 7 [17,23]. Thus, the time-dependent decrease in NADH fluorescence intensity is followed for 10 min at 450 nm (setting excitation at 366 nm) in the absence and presence of rotenone (Table 5). Complex I activity was determined by the difference between the slopes in the absence and presence of rotenone. A calibration curve of the NADH fluorescence intensity as a function of concentration, prepared in the same assay conditions, was used to convert the rate of NADH fluorescence decay promoted by Complex I activity in mol. Then, Complex I activity is expressed as nmol NADH oxidized per minute per mg of protein.

#### 2.3.2. Complex II: Succinate:Ubiquinone Oxidoreductase

Suspension of mitochondria-rich fraction (0.5 mg protein/mL);Phosphate buffer (KH_2_PO_4_ 25 mM, pH 7.5);2,6-dichlorophenolindophenol (DCPIP, 25 mM in water);KCN (100 mM in water);Antimycin A (2mM in ethanol);Decylubiquinone (38.75 mM in DMSO);Rotenone (600 µM in ethanol);Oxaloacetate (500 mM in water);Succinate (100 mM in water).

Complex II activity was evaluated spectrophotometrically following the reduction of DCPIP (color change from blue to colorless, Figure 8), which was used as an exogenous acceptor of the electrons. In assay conditions (i.e., Complex I is inhibited by rotenone, Complex III by antimycin A and Complex IV by KCN), succinate oxidation promoted by Complex II delivers the electrons for ubiquinone that promotes DCPIP reduction [17,24]. Thus, the time-dependent decrease in absorbance intensity at 600 nm (ε_DCPIP_ = 20,700 M^−1^ cm^−1^) in the absence and presence of oxaloacetate (competitive Complex II inhibitor [25]) is recorded in a microplate reader (Table 6). Complex II activity is determined by the difference between the slopes in the absence and presence of oxaloacetate. It is expressed as nmol DCPIP reduced per minute per milligram of protein. Considering the reactions in Complex II, the amount of DCPIP reduced is equal to the amount of fumarate formed by succinate oxidation. Thus, the activity can be expressed in nmol succinate oxidation per minute per milligram of protein.

#### 2.3.3. Complex IV: Cytochrome c Oxidase Activity

Suspension of mitochondria-rich fraction (0.5 mg protein/mL);Phosphate buffer (KH_2_PO_4_ 25 mM, pH 7.5);Antimycin A (2 mM in ethanol);Rotenone (600 µM in ethanol);KCN (100 mM in water);Reduced Cyt *c* (200 µM in phosphate buffer).

Complex IV activity was evaluated spectrophotometrically following the oxidation of reduced Cyt *c* (enzyme-substrate, ε_Cyt *c*_ = 29,500 M^−1^ cm^−1^ [26]) at 550 nm (color change from light red to dark red, Figure 9) [27]. Thus, the time-dependent decrease in absorbance intensity at 550 nm in the absence and presence of KCN (Complex IV inhibitor) is recorded for 10 min using a microplate reader, as indicated in Table 7. The stock solution of reduced Cyt *c* was freshly prepared (in a spectrophotometer cuvette, Helios Alpha, UNICAM) by adding a volume of dithionite saturated solution required to increase the absorbance of Cyt *c* solution in 0.6 units at 550 nm after its saturation of nitrogen gas. In general, 10µL of dithionite solution is enough to reduce 3 mL of a Cyt *c* solution at 200 µM. Complex IV activity was determined by the difference between the slopes in the absence and presence of KCN, and it is expressed as nmol oxidized Cyt *c* per minute per milligram of protein.

#### 2.3.4. Super-Complex I–III Activity

Suspension of mitochondria-rich fraction (0.5 mg protein/mL);Phosphate buffer (KH_2_PO_4_ 25 mM, MgCl_2_ 5 mM, pH 7.5);KCN (100 mM in water);Antimycin A (2 mM in ethanol);Rotenone (600 µM in ethanol);Oxidized Cyt *c* (200 µM in phosphate buffer);NADH (5 mM in water).

Super-complex I–III activity was evaluated spectrophotometrically following the reduction of oxidized Cyt *c* (enzyme substrate, ε_Cyt *c*_ = 29,500 M^−1^ cm^−1^, Figure 10) at 550 nm after activation of mitochondrial redox chain with NADH in the presence of KCN to inhibit Complex IV [27]. Thus, the time-dependent increase in absorbance intensity at 550 nm promoted by the mitochondria-rich fraction in the absence and presence of rotenone and antimycin (Complex I and III inhibitors, respectively) is recorded using a microplate reader (Table 8). Super-complex I–III activity is determined by the difference between the slopes in the absence and presence of inhibitors. It is expressed as nmol reduced Cyt *c* per minute per milligram of protein.

#### 2.3.5. Complex II–III Conjugated Activity

Suspension of mitochondria-rich fraction (0.5 mg protein/mL);Phosphate buffer (KH_2_PO_4_ 25 mM, pH 7.5);KCN (100 mM in water);Antimycin A (2 mM in ethanol);Rotenone (600 µM in ethanol);Oxidized Cyt *c* (200 µM in phosphate buffer);Succinate (100 mM in water).

Complex II–III conjugated activity is evaluated spectrophotometrically following the reduction of oxidized Cyt *c* (enzyme substrate, ε_Cyt *c*_ = 29,500 M^−1^ cm^−1^, Figure 11) at 550 nm after activation of the mitochondrial redox chain with succinate in the presence of rotenone and KCN to inhibit Complex I and Complex IV, respectively [27]. As indicated in Table 9, the time-dependent increase in absorbance intensity at 550nm promoted by the mitochondria-rich fraction in the absence and presence of antimycin (inhibitor Complex III) is recorded using a microplate reader. Complex II–III conjugated activity is determined by the difference between the slopes in the absence and presence of Complex III inhibitor. It is expressed as nmol reduced Cyt *c* per minute per milligram of protein.

#### 2.3.6. Citrate Synthase Activity

Suspension of mitochondria-rich fraction (0.5 mg protein/mL);Buffer (Tris-HCl 20 mM, pH 8, Triton X-100 0.02%);5,5-dithio-bis-2-nitrobenzoic acid (DTNB, 10 mM in water);Oxaloacetate (100 mM in water);Acetyl-CoA (6.1 mM in water);Succinyl-CoA (60 mM in water).

CS activity was determined spectrophotometrically, monitoring the color change associated with the reduction of DTNB by the CoASH, a product of the condensation reaction catalyzed by CS enzyme (acetyl-CoA plus oxaloacetate generate citrate and CoASH), as displayed in Figure 12 [17,28]. As indicated in Table 10, the time-dependent increase in absorbance at 412 nm is recorded in a microplate reader. Succinyl-CoA, an inhibitor of CS, is to confirm the reduction of DTNB in couple to CoASH released by the CS activity. CS activity is calculated by the difference between the slopes of DTNB reduction (ε_TNB_ = 13,600 M^−1^ cm^−1^) in the absence and presence of CS inhibitor. It is expressed as nmol TNB (like CoASH produced) per minute per milligram of protein.

## 3. Expected Results

### 3.1. Activities of the Cytosolic Antioxidant Enzymes 

In the cytosol, the Cu/Zn-SOD enzyme promotes the catalytic dismutation of O_2_^•−^ into O_2_ and H_2_O_2_. The generated H_2_O_2_ can be converted into H_2_O and O_2_ by CAT enzyme or processed by GPx enzyme under a catalytic cycle that uses the reducing power of GSH molecules to reduce H_2_O_2_ into H_2_O, accumulating oxidized glutathione (glutathione disulfide, GSSG). Since the GSH/GSSG redox pair is the most abundant redox couple in the cytosol and a key player in the homeostatic control of cell redox state, GPx works with the GR. This enzyme uses the reducing power of NADPH to regenerate GSH from GSSG.

Under the above-mentioned “sample” and “blank” conditions, the typical curves obtained for the activity of Cu/Zn-SOD, GR, GPx and CAT enzymes using 50 µg of protein mitochondria-free cytosolic fraction per assay, are shown in Figure 13A–D, respectively. The values of enzyme activities normalized by mg of protein, and the slopes of the linear functions obtained under “sample” and “blank” conditions used to calculate the corresponding enzyme activity, are displayed in Table 11. The activity of each enzyme (mean ± std) of three independent assays performed with mitochondria-free cytosolic fraction obtained from cells of three consecutive passages were also included in Table 11 to show the reproducibility of the methodologies.

For GR, GPx and CAT, the values of the enzymes’ activity presented in Table 11 represent the effective amount of substrate (GR, GPx) or product (CAT) that one mg of sample protein can transform or produce for one minute. Thereby the enzyme activities can be compared among them for the same sample. On the other hand, the activity of Cu/Zn-SOD enzyme depends on the xanthine/xanthine oxidase system used for O_2_^•−^ generation and on the NBT concentration used for detection. Therefore, the Cu/Zn-SOD activity values have higher relevance for comparison among samples obtained with cells subjected to different stimuli than with the activity of the other antioxidant enzymes in the sample. Despite this, Table 11 indicates that the mitochondria-free cytosolic fraction of SH-SY5Y cells exhibits high Cu/Zn-SOD enzyme activity, at least an order of magnitude greater than the remaining antioxidant enzymes. Additionally, the activity of CAT is significantly higher than that of GR and GPx enzymes. Compared with GPx, the sample exhibit higher activity of GR, suggesting that under the cell growth conditions tested, cells can use GPx to handle the H_2_O_2_ preserving the GSH/GSSG ratio and the homeostatic control on cytosolic redox state. It is also important to highlight that the sample has a higher activity of CAT than GPx, two enzymes involved in the decomposition of H_2_O_2_. However, as observed in the experimental section, CAT requires a higher substrate concentration to reach the maximum velocity. Thus, the GPx should be the most relevant enzyme to handle the low rates of H_2_O_2_ generated under physiological conditions. However, the stage is given to CAT under toxic stimuli that promote oxidative burst.

### 3.2. Mitochondrial Respiratory Chain Complexes Activities

The mitochondrial respiratory chain is activated at the level of Complex I by the electrons delivered by NADH and at the level of Complex II by FADH_2_ resulting from succinate oxidation. The ubiquinone pool accepts the electrons from Complex I or Complex II. It delivers them to Complex III, reducing Cyt *c* that transports electrons for Complex IV, where the O_2_ is reduced to water. According to the chemiosmotic hypothesis [29], the energy of oxidations is transduced into a transmembrane proton gradient generating a transmembrane electric potential used for many functions, including ATP synthesis. In this way, the citric acid cycle in the mitochondrial matrix, and the mitochondrial respiratory chain (in the inner membrane) are interlinked processes connecting the carbon fluxes with the membrane electron fluxes and ATP synthesis. Thus, measuring mitochondrial respiratory chain complexes’ activities is essential to investigate mitochondrial function in physiological and pathological conditions, including neurodegenerative diseases [30]. 

The typical curves obtained for the Complexes I, II and IV, the conjugated activity of Complexes I–III and II–III and CS enzyme, under the above-mentioned “sample” and “sample with an inhibitor” conditions using 10 µg of protein mitochondria-rich fraction per assay, are shown in Figure 14A–F panels, respectively. The mitochondrial respiratory chain complex activities are calculated by the difference between the slopes of the linear functions obtained under “sample” and “sample with inhibitor” conditions. The obtained values normalized by mg of protein and by the activity of CS are displayed in Table 12. The activity of each enzyme (mean ± std) of three independent assays performed with mitochondria-rich fraction obtained from cells of three consecutive passages were also included in Table 12 to show the reproducibility of the methodologies. 

In line with the theoretically expected the activity of Complex I is about three times higher than Complex II activity since in each turn of the citric acid cycle are produced 3 molecules of NADH and only one of FADH_2_. Although the direct activity of Complex III was not evaluated, the data in Table 12 also suggest that Complex IV limits the mitochondrial respiratory chain activity in SH-SY5Y cells grown under the above-described conditions. Data from Table 12 also suggest that the capacity of the citric acid cycle to produce NADH exceeds respiratory chain processing capacity. Thus, a significant part of the reduced co-enzyme and other citric acid cycle intermediates can be used to support the biosynthetic requirements of the SH-SY5Y cells in a proliferative state. CS is generally considered a rate-limiting citric acid cycle enzyme [31,32]. However, the activity of Complex II, which is also an enzyme of the citric acid cycle, is significantly lower than the CS activity, suggesting that succinate oxidation can be a rate-limiting step of the citric acid cycle in SH-SY5Y cells.

### 3.3. Relationship between Mitochondrial Redox Chain Functionality and the Cytosolic Antioxidant Enzymes Activity

The functional organization of respiratory chain components in the inner mitochondrial membrane, the pressure of electron flux along the mitochondrial respiratory chain, and the intracellular oxygen availability determine the production of ROS, mainly as O_2_^•−^/H_2_O_2_. Experimental data indicate that in many pathological conditions, including those that mimic the neurodegenerative process with mitochondrial dysfunction, the ROS generation increases above the range of values recorded in paired physiological conditions [8,33,34]. Moreover, Complex I is considered the primary source of mitochondrial ROS generation. The process is mainly explained by a one-electron redox event, involving the transfer of one electron from the iron-sulfur centers of Complexes I to the molecular oxygen. In fact, the mitochondrial ROS generation have been also associated to complex III, and the functioning of this complex also involves one-electron redox events [35]. Thus, the O_2_^•−^ is the main ROS released by mitochondria [4,5,6,33]. In fact, Complex I have the highest activity compared to the other complexes’ activities, indicating that its capacity to deliver electrons exceeds the capacity of other respiratory complexes, particularly Complex IV, to receive them. Thus, the mitochondrial Complex I of SH-SY5Y cells have a high potential to release O_2_^•−^. Despite this, cellular oxidative damage only occurs if and when the ROS generation overpasses the capacity of the cellular antioxidant defense system to handle them. Consequently, assessing the activity of antioxidant enzymes that directly (SOD) and indirectly (CAT, GPX and GR) deal with the superoxide radical anion is necessary to establish the relationship between the mitochondrial redox chain functionality and oxidative stress or cell redox state. Thus, it is also not surprising that in the cytosol of SH-SY5Y cells, the activity of Cu/Zn-SOD is one order of magnitude higher than the other antioxidant enzymes.

## 4. Conclusions

This article describes an experimental approach to evaluate the putative relationship between mitochondrial dysfunction and cytosolic oxidative stress in cell culture systems grown in a single T-25 or T-75 flasks. It provides detailed procedures to promote cell fractionation by differential centrifugation and to assess: (i) in mitochondria-rich fraction, the activity of the mitochondrial respiratory chain complexes (Complex I, II, IV, super-complex I–III and Complex II–III conjugated activity) normalized by the activity of CS; (ii) in the mitochondria-free cytosolic fraction, the activities of the main cytosolic antioxidant enzymes (Cu/Zn-SOD, CAT, GR and GPx). Despite the previous articles with the procedures for mitochondria complex activity assessment [36,37], the information obtained with the current approach is distinct, addressing the relationship between oxidative stress/cytosolic redox state and mitochondrial dysfunction. Tools that can be used to understand the response of cells to oxidant stimuli (or other stress agents) as well as to compare different types of cells under the control conditions. Additionally, the present methodology can be used to complement and/or explain data obtained with other methodologies, including the intracellular ROS generation (usually detected with fluorescent probes such as dichloro-dihydro-fluorescein diacetate), the mitochondrial membrane potential (usually detected with mitochondriotropic fluorescent probes such as tetraethylbenzimidazolocarbocyanine iodide), and the mitochondrial performance considering the extracellular rates of oxygen fluxes (e.g., Seahorse technology) [10,38,39,40]. Standardized protocols also allow the comparison of results among research laboratories. They can be used in drug research programs to improve the translation of in vitro findings to in vivo studies. 

## Figures and Tables

**Figure 1 mps-06-00032-f001:**
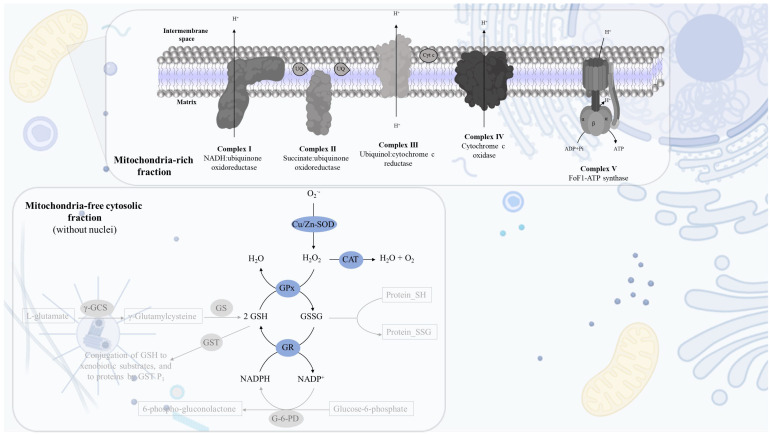
Schematic representations of the electron respiratory chain and the antioxidant defense system within the cell. The mitochondrial redox chain is composed of Complex I (NADH:ubiquinone oxidoreductase), Complex II (succinate:ubiquinone oxidoreductase), Complex III (ubiquinol:cytochrome c reductase), Complex IV (cytochrome c oxidase) and Complex V (FoF1-ATP synthase) in the inner mitochondrial membrane on cristae. Antioxidant enzymes mainly: Cu/Zn-SOD that is responsible for catalyzing the dismutation of O_2_^•−^ into H_2_O_2_ and O_2_; CAT promotes the conversion of H_2_O_2_ into H_2_O and O_2_; GPx reduces the H_2_O_2_ into H_2_O oxidizing two molecules of GSH to form an oxidized dimer GSSG; and GR that uses the reducing power of NADPH to regenerate GSH from GSSG. ADP—adenosine diphosphate, ATP—adenosine triphosphate, CAT—catalase, Cu/Zn-SOD—superoxide dismutase, Cyt *c*—cytochrome *c*, GPx—glutathione peroxidase, GR—glutathione reductase, GS—glutathione synthetase, GSH—glutathione reduced, GSSG—glutathione oxidized, GST—glutathione S-transferases, G-6-PD—glucose-6-phosphate dehydrogenase, H_2_O—water, H_2_O_2_—hydrogen peroxide, NADPH/NADP^+^—nicotinamide adenine dinucleotide phosphate, O_2_—molecular oxygen, O_2_^•−^—superoxide, UQ—ubiquinone, γ-GCS—gamma-glutamylcysteine synthetase.

**Figure 2 mps-06-00032-f002:**
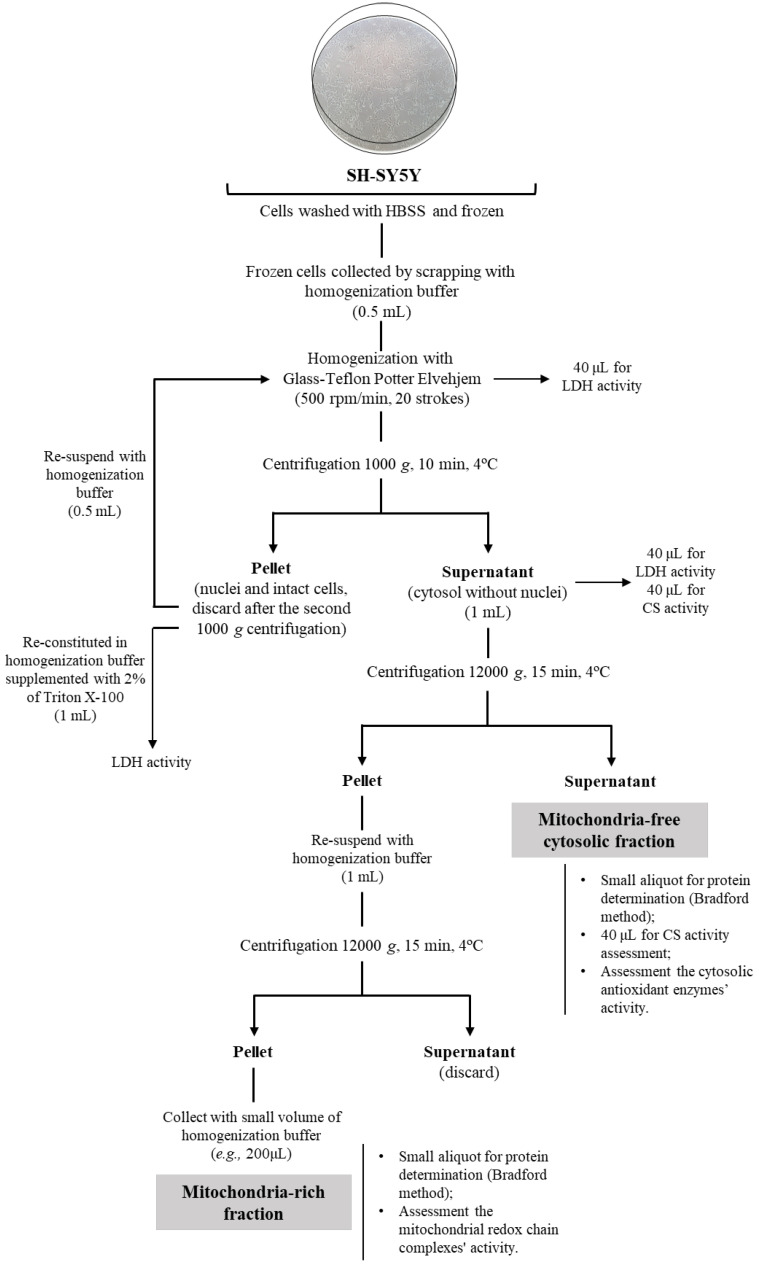
Flow diagram representing the fractionation of cell pellet in mitochondria-free cytosolic and mitochondria-rich fractions on cell cultures by differential centrifugation.

**Figure 3 mps-06-00032-f003:**
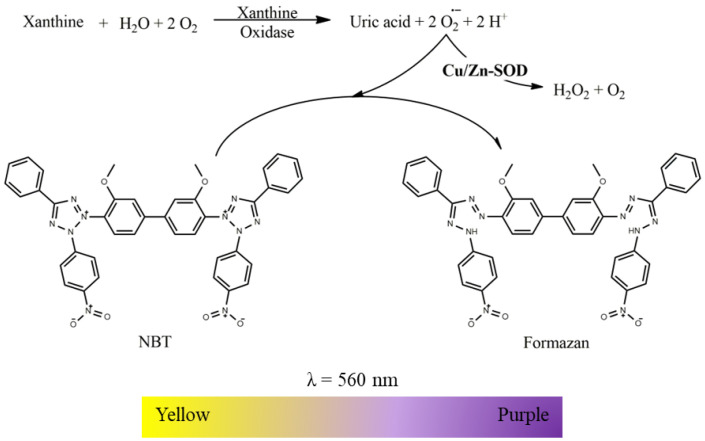
The xanthine/xanthine oxidase system generates O_2_^•−^ disputed between Cu/Zn-SOD enzyme and the yellow NBT. Cu/Zn-SOD is responsible for the dismutation of O_2_^•−^ into H_2_O_2_ and molecular oxygen, and the NBT reacts with the O_2_^•−^ producing the formazan, detectable at 560 nm.

**Figure 4 mps-06-00032-f004:**
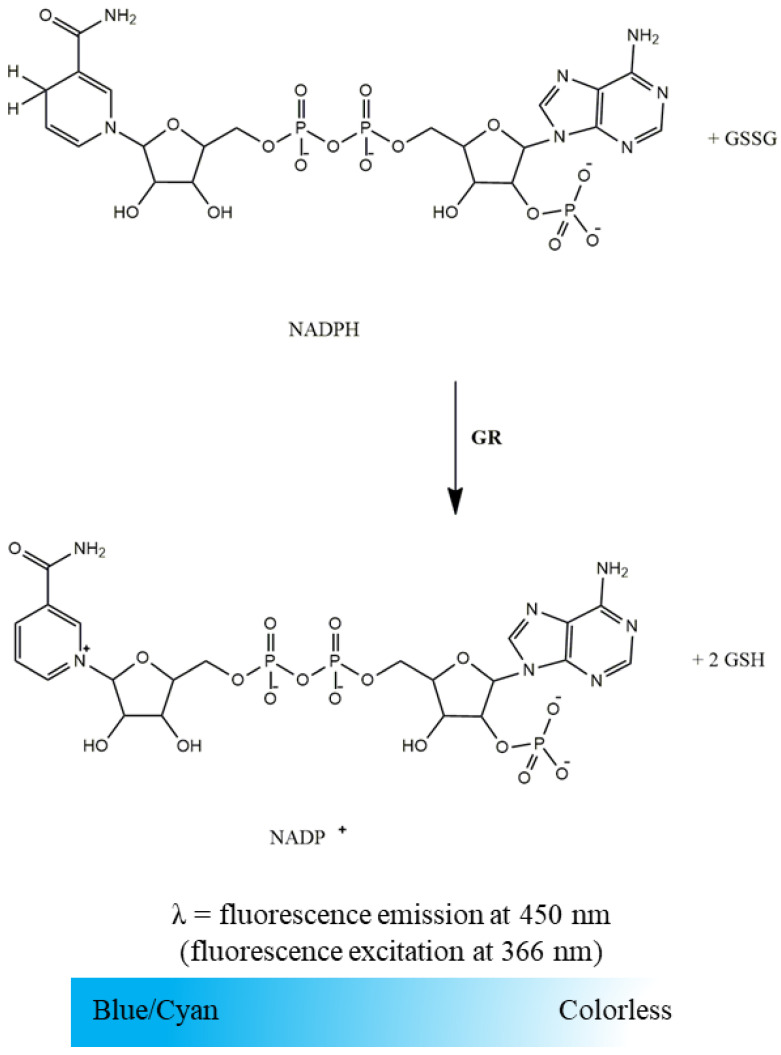
Oxidation of NADPH to NADP^+^ with reduction of GSSG to GSH by GR. The NADPH oxidation intensity is measured at 450 nm (setting excitation at 366 nm).

**Figure 5 mps-06-00032-f005:**
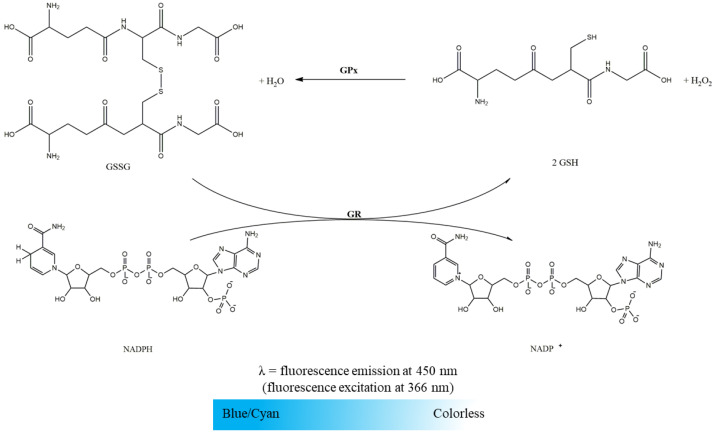
GPx activity is evaluated by a coupled reaction with GR enzyme. GPx reduces hydrogen peroxide to water and oxidizes GSH to GSSG. The generated GSSG is reduced to GSH with NADPH consumption by GR.

**Figure 6 mps-06-00032-f006:**
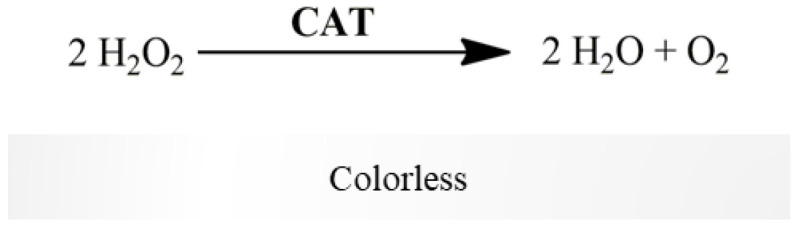
CAT reduces H_2_O_2_ to water and O_2_, detected using a Clark-type oxygen electrode.

**Figure 7 mps-06-00032-f007:**
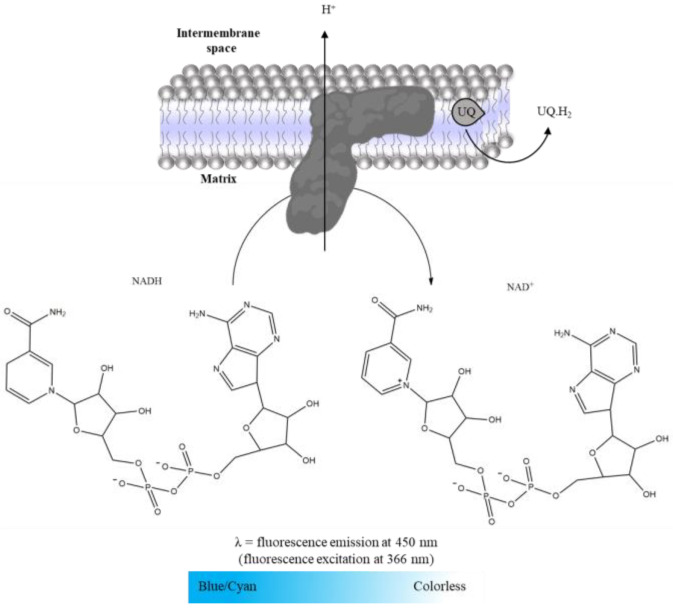
Oxidation of NADH to NAD+, measured at 450nm (setting excitation at 366 nm).

**Figure 8 mps-06-00032-f008:**
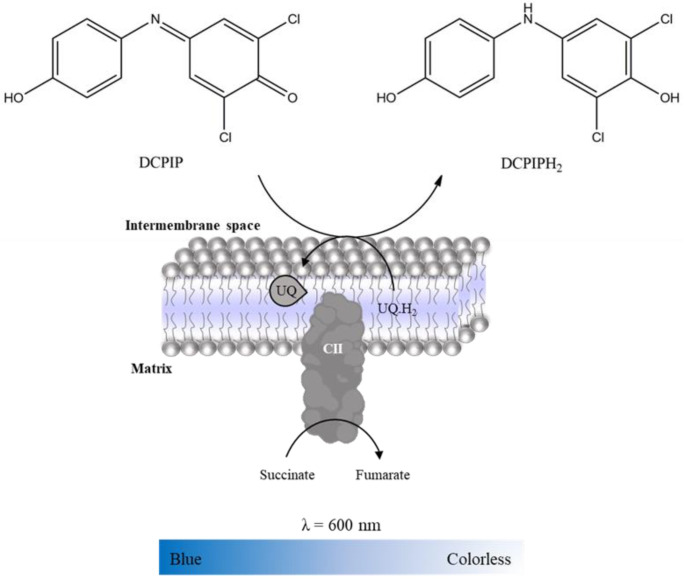
Reduction of DCPIP with color change from blue to colorless, used as an exogenous acceptor of electrons resulting from succinate oxidation to fumarate promoted by Complex II.

**Figure 9 mps-06-00032-f009:**
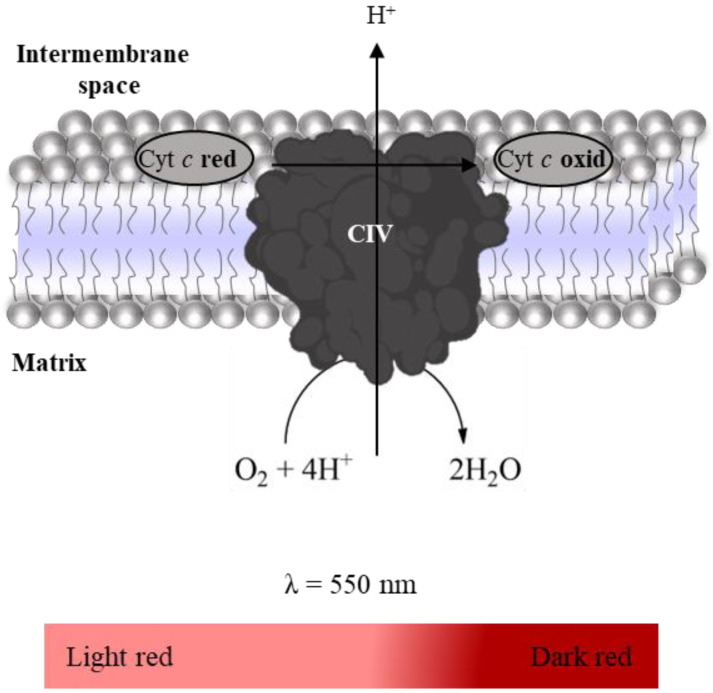
Oxidation of the reduced Cyt *c* with water production.

**Figure 10 mps-06-00032-f010:**
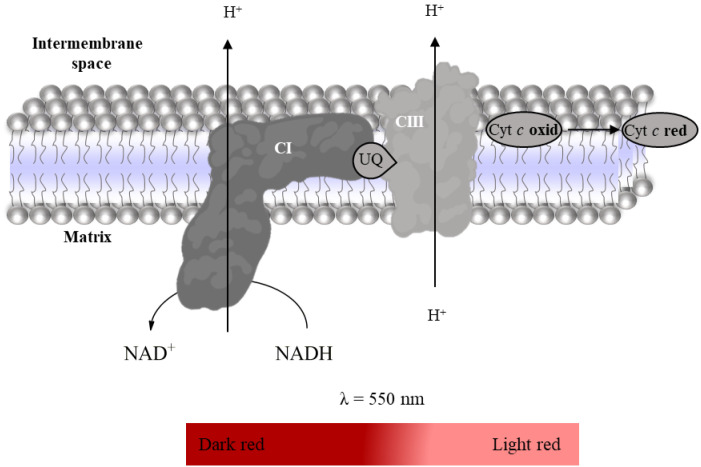
Reduction of oxidized Cyt *c* with oxidation of NADH to NAD+.

**Figure 11 mps-06-00032-f011:**
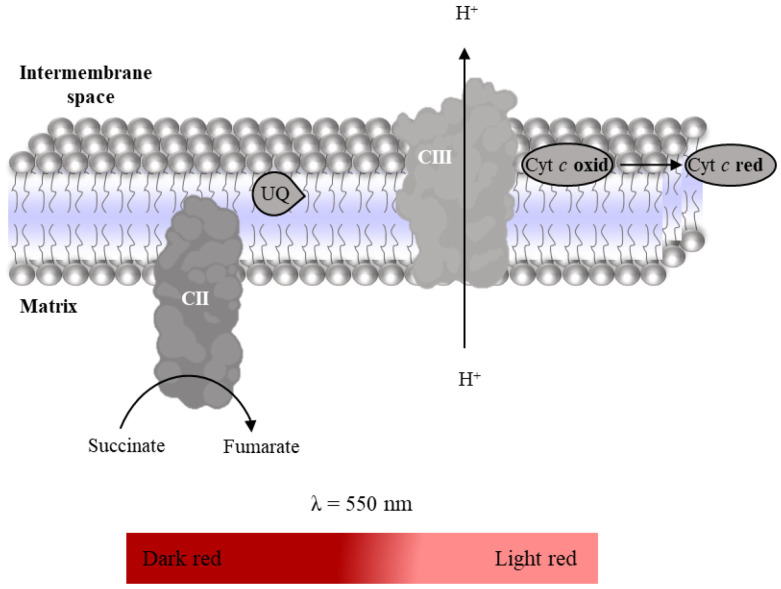
Reduction of oxidized Cyt *c* with succinate oxidation to fumarate.

**Figure 12 mps-06-00032-f012:**
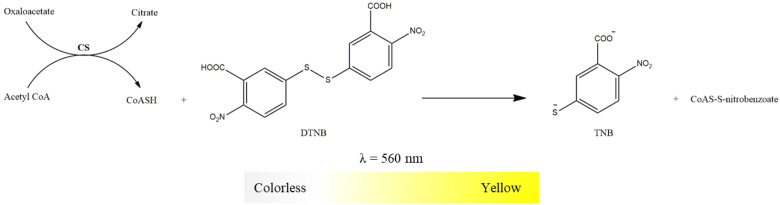
Reduction of DTNB with color change from colorless to yellow by the CoASH resulting from the condensation reaction of acetyl-CoA and oxaloacetate to form citrate promoted by CS.

**Figure 13 mps-06-00032-f013:**
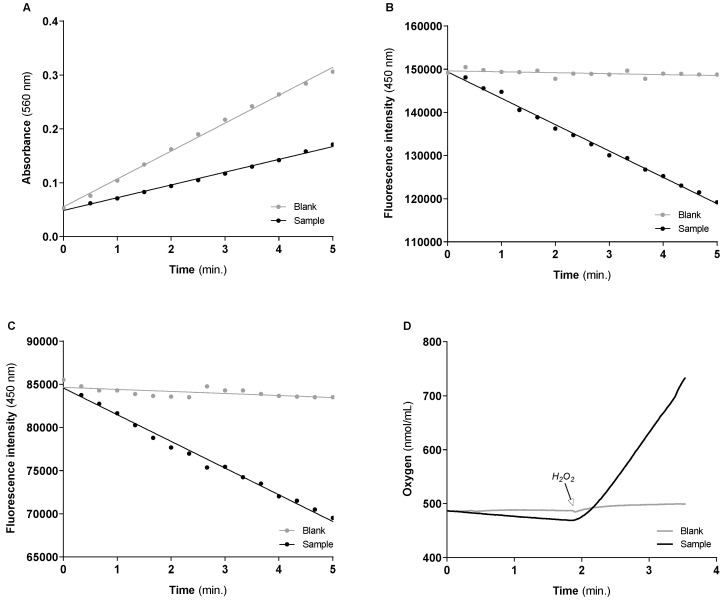
Typical progress curves were used to assess the activities of Cu/Zn-SOD (**A**), GR (**B**), GPx (**C**) and CAT (**D**) enzymes in the mitochondria-free cytosolic fraction. Enzyme activity is calculated by the difference (or module of the difference) between the slopes of the black lines (sample) and grey lines (blank). The “blank” conditions represent in: (**A**), the detection of the O_2_^•−^ produced by xanthine/xanthine oxidase system in the absence of mitochondria-free cytosolic fraction; (**B**), NADPH oxidation in the absence of GSSG; (**C**), NADPH oxidation in the absence of GSH and presence of GR enzyme; (**D**), oxygen production induced by H_2_O_2_ addition in the absence of the biological sample.

**Figure 14 mps-06-00032-f014:**
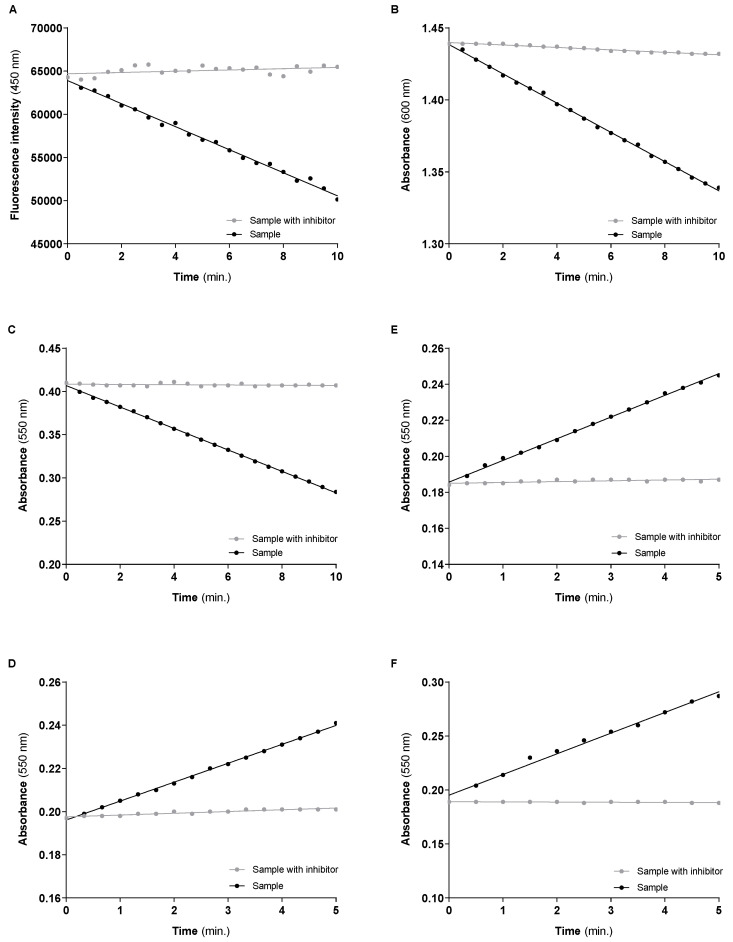
Typical progress curves were used to assess the activities of Complex I (**A**), II (**B**), IV (**C**), I–III (**D**), II–III (**E**) and CS (**F**) in the mitochondria-rich fraction. The mitochondrial respiratory chain complexes activities and the CS activity are calculated by the difference between the slopes on the presence (grey lines) and absence (black lines) of the inhibitors used for each activity.

**Table 1 mps-06-00032-t001:** Conditions and flow setup for Cu/Zn-SOD activity assay.

	Sample	Blank
	Volume (μL)	Volume (μL)
**Buffer**	130	180
**Mitochondria-free cytosolic fraction**	50	0
**NBT**	20	20
**Xanthine oxidase**	10	10
Incubate for 5 min, 37 °C		
**Xanthine**	40	40
**Read absorbance** (560 nm)	Kinetics for 5 min, recording points every 30 s	

**Table 2 mps-06-00032-t002:** Conditions and flow setup for GR activity assay.

	Sample	Blank
	Volume (μL)	Volume (μL)
**Buffer**	185	195
**Mitochondria-free cytosolic fraction**	50	50
**GSSG**	10	0
Incubate for 5 min, 37 °C		
**NADPH**	5	5
**Read fluorescence** (Ex 366 nm; Em 450 nm)	Kinetics for 5 min, recording points every 20 s	

**Table 3 mps-06-00032-t003:** Conditions and flow setup for GPx activity assay.

	Sample	Blank
	Volume (μL)	Volume (μL)
**Buffer**	155	165
**Mitochondria-free cytosolic fraction**	50	50
**GSH**	10	0
**GR**	10	10
**NADPH**	5	5
Incubate for 5 min, 37 °C		
**H_2_O_2_**	20	20
**Read fluorescence** (Ex 366 nm; Em 450 nm)	Kinetics for 5 min, recording points every 20 s	

**Table 4 mps-06-00032-t004:** Conditions and flow setup for CAT activity assay.

	Sample	Blank
	Volume (μL)	Volume (μL)
**Buffer**	930	980
**Mitochondria-free cytosolic fraction**	50	0
Incubate for 5 min, 37 °C		
**H_2_O_2_**	20	20
**Read oxygen production**	Kinetics for 2–3 min, recording points every 1 s	

**Table 5 mps-06-00032-t005:** Conditions and flow setup for Complex I activity assay.

	Sample	Sample with Inhibitor
	Volume (μL)	Volume (μL)
**Buffer**	214	209
**Mitochondria-rich fraction**	20	20
**KCN**	2.5	2.5
**Antimycin A**	1	1
**Decylubiquinone**	5	5
**Rotenone**	0	5
Incubate for 5 min, 37 °C		
**NADH**	7.5	7.5
**Read fluorescence** (Ex 366 nm; Em 450 nm)	Kinetics for 10 min, recording points every 30 s	

**Table 6 mps-06-00032-t006:** Conditions and flow setup for Complex II activity assay.

	Sample	Sample with Inhibitor
	Volume (μL)	Volume (μL)
**Buffer**	212.75	207.75
**Mitochondria-rich fraction**	20	20
**DCPIP**	2.5	2.5
**KCN**	2.5	2.5
**Antimycin A**	1	1
**Decylubiquinone**	1.25	1.25
**Rotenone**	5	5
**Oxaloacetate**	0	5
Incubate for 5 min, 37 °C		
**Succinate**	5	5
**Read absorbance** (600 nm)	Kinetics for 10 min, recording points every 30 s	

**Table 7 mps-06-00032-t007:** Conditions and flow setup for Complex IV activity assay.

	Sample	Sample with Inhibitor
	Volume (μL)	Volume (μL)
**Buffer**	189	186.5
**Mitochondria-rich fraction**	20	20
**Antimycin A**	1	1
**Rotenone**	5	5
**KCN**	0	2.5
Incubate for 5 min, 37 °C		
**Reduced Cyt *c***	35	35
**Read absorbance** (550 nm)	Kinetics for 10 min, recording points every 30 s	

**Table 8 mps-06-00032-t008:** Conditions and flow setup for Super-complex I–III activity assay.

	Sample	Sample with Inhibitor
	Volume (μL)	Volume (μL)
**Buffer**	202.5	196.5
**Mitochondria-rich fraction**	20	20
**KCN**	2.5	2.5
**Antimycin A**	0	1
**Rotenone**	0	5
**Oxidized Cyt *c***	20	20
Incubate for 5 min, 37 °C		
**NADH**	5	5
**Read absorbance** (550 nm)	Kinetics for 5 min, recording points every 30 s	

**Table 9 mps-06-00032-t009:** Conditions and flow setup for Complex II–III conjugated activity assay.

	Sample	Sample with Inhibitor
	Volume (μL)	Volume (μL)
**Buffer**	197.5	196.5
**Mitochondria-rich fraction**	20	20
**KCN**	2.5	2.5
**Antimycin A**	0	1
**Rotenone**	5	5
**Oxidized Cyt *c***	20	20
Incubate for 5 min, 37 °C		
**Succinate**	5	5
**Read absorbance** (550 nm)	Kinetics for 5 min, recording points every 30 s	

**Table 10 mps-06-00032-t010:** Conditions and flow setup for CS activity assay.

	Sample	Sample with Inhibitor
	Volume (μL)	Volume (μL)
**Buffer**	210	205
**Mitochondria-rich fraction**	20	20
**DTNB**	10	10
**Oxaloacetate**	5	5
**Succinyl-CoA**	0	5
Incubate for 5 min, 37 °C		
**Acetyl-CoA**	5	5
**Read absorbance** (412 nm)	Kinetics for 5 min, recording points every 30 s	

**Table 11 mps-06-00032-t011:** Typical linear regressions of cytosolic antioxidant enzymes obtained in assays with 50 μg of protein under “blank” and “sample” conditions (Figure 13) and the calculated enzyme activity.

		Linear Regression	Enzyme Activity
**Cu/Zn-SOD**	Blank	Y = 0.05178 × X + 0.05527	9494 nmol/min/mg of protein (9051.509 ± 377.986) ^#^
	Sample	Y = 0.02731 × X + 0.04855	
**GR**	Blank	Y = −216.9 × X + 149,618	47.486 nmol NADPH/min/mg of protein (47.572 ± 3.324) ^#^
	Sample	Y = −6108 × X + 149,379	
**GPx**	Blank	Y = −236.9 × X + 84,650	23.038 nmol H_2_O_2_/min/mg of protein (24.459 ± 1.083) ^#^
	Sample	Y = −3095 × X + 84,577	
**CAT**	Blank	Y = 2.835 × X + 446.5	560.217 nmol O_2_/min/mg of protein (583.549 ± 16.954) ^#^
	Sample	Y = 170.9 × X + 120.0	

^#^ Mean ± std of three independent assays

**Table 12 mps-06-00032-t012:** Typical linear regressions for mitochondrial respiratory chain complexes were obtained in assays with 10 μg of protein for “sample” and “sample with inhibitor” conditions, the enzyme activity normalized by mg of protein and the activity of each complex normalized by the CS activity.

		Linear Regression	Enzyme Activity	Enzyme Activity/CS
**CI**	Sample	Y = −1334 × X + 63,904	54.629 nmol NADH/min/mg of protein (55.485 ± 4.313) ^#^	0.939
	Sample w/inhibitor	Y = 73.32 × X + 64,694		
**CII**	Sample	Y = −0.01016 × X + 1.438	18.736 nmol DCPIP reduced/min/mg of protein * (20.722 ± 1.796) ^#^	0.322
	Sample w/inhibitor	Y = −0.0008519 × X + 1.440		
**CIV**	Sample	Y = −0.01242 × X + 0.4068	17.319 nmol oxidized Cyt c/min/mg of protein (18.786 ± 1.181) ^#^	0.298
	Sample w/inhibitor	Y = −0.0001584 × X + 0.4085		
**CI–III**	Sample	Y = 0.01205 × X + 0.1856	16.372 nmol reduced Cyt c/min/mg of protein (18.808 ± 2.254) ^#^	0.281
	Sample w/inhibitor	Y = 0.0004588 × X + 0.1856		
**CII–III**	Sample	Y = 0.008762 × X + 0.1961	11.229 nmol reduced Cyt c/min/mg of protein (9.912 ± 1.011) ^#^	0.193
	Sample w/inhibitor	Y = 0.0008118 × X + 0.1976		
**CS**	Sample	Y = 0.01916 × X + 0.1952	58.200 nmol TNB/min/mg of protein ** (63.204 ± 5.699) ^#^	-
	Sample w/inhibitor	Y = −0.0001636 × X + 0.1891		

* Considering the combined reactions, CII activity is 18.736 nmol succinate oxidized/min/mg of protein. ** Considering the combined reactions, CS activity is 58.200 nmol acetyl-CoA/min/mg of protein. ^#^ Mean ± std of three independent assays.

## Data Availability

Not applicable.

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
