# Peer review of "An Experimental Approach to Address the Functional Relationship between Antioxidant Enzymes and Mitochondrial Respiratory Complexes"

_mps, 2023, doi:10.3390/mps6020032_

Round 1
Reviewer 1 Report
The data presented in the article evaluate a number of parameters linking ETC enzymes to ROS production with antioxidant defense enzymes.
In the article, I did not find new techniques for isolating different cell fractions. As for the determination of enzyme activities, the article presents modifications of the main known methods for enzyme determination, but I am not sure if they are successful.
Measurement of ETC enzyme activity is well described in an article by Marco Spinazzi et al. (Assessment of mitochondrial respiratory chain enzymatic activities on tissues and cultured cells., doi:10.1038/nprot.2012.058).
I understand that the amount of mitochondrial fraction obtained from cells is very small, but to assess the development of oxidative stress, it would be good to measure the activity of Mn-SOD, catalase and glutathione peroxidase exactly in this fraction. Especially the activity of Cu-Zn SOD, may differ from Mn-SOD, which is localized in mitochondria. This is also true for the other antioxidant enzymes.
Nevertheless, I believe that the article can be published and will find its experimenter depending on the goals and possibilities.
There are a few notes on the text:
Line 56. Mitochondria-free cytosolic faction is used to assess the activity of some.... There is a typo here.
Did you use anything other than a homogenizer (catalog number, homogenizer clearance?) to disrupt the cell membrane? How did you check? Is the use of a single homogenizer enough to destroy the cells?
Glutathione peroxidase and glutathione reductase activities are usually perfectly measured on a spectrophotometer at 340 nm and catalase at 240 nm. What are the advantages of measuring the activities of these enzymes spectrofluorimetrically and catalase polarographically? What is the maximum concentration of H2O2 that can be measured polarographically? Is there a calibration curve for H2O2 for the Clark electrode?
Is it necessary to incubate Mitochondria-free cytosolic fraction for 5 min at 37°C?
You are using oxaloacetate as an SDH inhibitor, there are more established inhibitors of this enzyme: TTFA, malonate. Why oxaloacetate? Have you done a concentration dependence curve of enzyme inhibition with oxaloacetate?
The molar extinction coefficient of reduced cytochrome c is 29500M-1*cm-1. Please give a reference to this value in the literature, since a value of 18500 is also found.
2.3.4. Super-complex I-III activity. Do you count Complex I-III activity as a super-complex here?
2.3.5. Complex II-III conjugated activity, why Complex II-III conjugated activity.
Author Response
Comment 1: The data presented in the article evaluate a number of parameters linking ETC enzymes to ROS production with antioxidant defense enzymes. In the article, I did not find new techniques for isolating different cell fractions. As for the determination of enzyme activities, the article presents modifications of the main known methods for enzyme determination, but I am not sure if they are successful. Measurement of ETC enzyme activity is well described in an article by Marco Spinazzi et al. (Assessment of mitochondrial respiratory chain enzymatic activities on tissues and cultured cells., doi:10.1038/nprot.2012.058). I understand that the amount of mitochondrial fraction obtained from cells is very small, but to assess the development of oxidative stress, it would be good to measure the activity of Mn-SOD, catalase and glutathione peroxidase exactly in this fraction. Especially the activity of Cu-Zn SOD, may differ from Mn-SOD, which is localized in mitochondria. This is also true for the other antioxidant enzymes. Nevertheless, I believe that the article can be published and will find its experimenter depending on the goals and possibilities. There are a few notes on the text: Line 56. Mitochondria-free cytosolic faction is used to assess the activity of some.... There is a typo here.
Answer to comment 1: We agree with the relevance of the reviewer's comment. Firstly, we confirm that the cell fractionation procedure used to obtain the mitochondria-free cytosolic fraction and mitochondria-rich fraction are the routine procedures that we use for cell and animal tissues, as previously reported (References 22, 29). Secondly, we highlight that the evaluation of the activity of antioxidant enzymes in mitochondria-rich fractions is an important issue since mitochondrion has its own pool of antioxidant enzymes, including the Mn-SOD - an isoform of SOD enzyme that does not occur in the cytosol. While the described procedures to assess the activity of antioxidant enzymes in the cytosolic fraction are also suitable for mitochondria-rich fraction, the amount of mitochondria obtained with a T25 or T75 flasks do not allow the measurement of both the respiratory complexes and antioxidant enzyme activities. Thus, we built the manuscript with a strategy to evaluate the relationship between antioxidant enzymes in the cytosol and mitochondrial respiratory complexes' activity, maximizing the use of the available biological samples collected in one flask. The typo in line 56 was corrected.
Comment 2: Did you use anything other than a homogenizer (catalog number, homogenizer clearance?) to disrupt the cell membrane? How did you check? Is the use of a single homogenizer enough to destroy the cells?
Answer to comment 2: They are interesting questions. Liquid homogenization with a Potter-Elvehjem homogenizer is a widely used methodology to promote cell disruption of small volumes of cell suspensions and soft tissues (e.g., brain and liver), preserving the integrity and functionality of the intracellular organelles. In this methodology, cells are pushed through a narrow space between the intern wall of an unground rounded-shaped glass vessel and a PTFE pestle, and shear forces are used to disrupt cell membranes, producing cell lysates. In the present work, cell suspensions were homogenized in a 3 mL Potter-Elvehjem homogenizer (Merck, Reference P7734-1EA; clearance of 0.004 - 0.006 in.) using 20 up- and downstrokes at 500 rpm. The efficacy of cell lysis was evaluated by measuring the lactate dehydrogenase activity in cell homogenate, cell pellet reconstituted in 1 mL of buffer solution supplemented with 2% of Triton X-100, and in the supernatant fraction. Figure 2 was changed to include this information. Using SH-SY5Y cells, we can also assert that the LDH activity is not detected in 40 µL of the reconstituted pellet and more than 95% of this enzyme activity exhibited by cell homogenate is preserved in the supernatant fraction. Thus, the procedure is suitable to promote cell lysis. This information is also included in the new version of the manuscript.
Comment 3: Glutathione peroxidase and glutathione reductase activities are usually perfectly measured on a spectrophotometer at 340 nm and catalase at 240 nm. What are the advantages of measuring the activities of these enzymes spectrofluorimetrically and catalase polarographically? What is the maximum concentration of H2O2 that can be measured polarographically? Is there a calibration curve for H2O2 for the Clark electrode?
Answer to comment 3: Regarding glutathione peroxidase and glutathione reductase activities, the NADPH decay can be followed by using a spectrophotometer at 340 nm or by using a spectrofluorometer setting the emission at 450 nm and excitation at 366 nm. The fluorescence-based methodology was selected since, under the above-mentioned conditions, NADPH is a strong fluorophore while its oxidized pair (NADP+) is non-fluorescent, and for analysis of biological samples, in general, the fluorescence-based quantification is more sensitive than UV-based absorption measurement [https://doi.org/10.1007/978-0-387-46312-4_19]. This explanation is included in the manuscript.
Regarding the catalase activity assay, our option was to evaluate the activity of this enzyme recording the changes in oxygen concentration in the reaction medium to avoid the drawbacks associated with the measurements of the substrate (H2O2) concentration decay by the time-dependent changes in the absorption at 240 nm. In fact, biological samples exhibit strong absorption at 240 nm resulting from the presence of lipids, proteins, and nucleotides, producing strong levels of background with a consequent decrease of the signal/noise ratio, hindering the detection of small changes in H2O2 levels commons in biological samples with low levels of catalase. As previously described, measuring the activities of catalase polarographically is considered an accurate method, and compatible with complex biological samples, avoiding the above-mentioned matrix-related drawbacks (https://doi.org/10.1016/0003-2697(77)90662-5; https://doi.org/10.3390/oxygen3010006 ). Additionally, catalase and glutathione peroxidase have H2O2 as substrate, but only catalase generate oxygen. Thus, the assessment of catalase activity by oxygen generated avoid the use of a glutathione peroxidase inhibitor in the reaction medium.
Regarding the questions “What is the maximum concentration of H2O2 that can be measured polarographically? Is there a calibration curve for H2O2 for the Clark electrode?”, it is important to highlight that all assays were designed to assess the amount of enzyme active in the biological samples. Thus, the assay condition for each enzyme requires that the substrate concentration in reaction medium ensure that the enzyme works at Vmax, at least, for two minutes, i.e., the amount of enzyme in biological samples is the rate limiting factor of the reaction. Hansatech Oxygraph Instrument allows to detect at each second changes in oxygen concentration of 0.01 nmol/mL range. However, under our catalase assay conditions, the 10-fold decrease in the biological sample (i.e., to 5 mg protein/mL) still allow reliably detection of catalase activity, as shown in the supplementary Figure S1 (O2 production rate = 22.92 nmol/mL/min., corresponding to H2O2 consumption rate of 45.84 nmol/mL/min according with stoichiometric of the catalase-catalysed reaction). This information was included in new version of manuscript.
Comment 4: Is it necessary to incubate Mitochondria-free cytosolic fraction for 5 min at 37°C?
Answer to comment 4: The incubation of biological samples at 37 ºC for 5 min before starts the reaction is an operational strategy to ensure that the activity of the enzyme is assessed at physiological temperature, 37 ºC for humans.
Comment 5: You are using oxaloacetate as an SDH inhibitor, there are more established inhibitors of this enzyme: TTFA, malonate. Why oxaloacetate? Have you done a concentration dependence curve of enzyme inhibition with oxaloacetate?
Answer to comment 5: The oxaloacetate, a competitive inhibitor of the Complex II with a strong affinity for succinate-binding site of the enzyme as indicated by the constant inhibition of 1.5 μM (10.1104/pp.70.6.1577), is a suitable for assay proposal. The objective of the work is not to characterize the enzyme kinetics inhibition but to assess the activity of the enzyme in the biological samples. We can also assert that the use of malonate (another competitive inhibitor) and TTFA, a non-competitive inhibitor, are also suitable for assay.
Comment 6: The molar extinction coefficient of reduced cytochrome c is 29500M-1*cm-1. Please give a reference to this value in the literature, since a value of 18500 is also found.
Answer to comment 6:
29500M-1*cm-1 is used in several articles as value of the molar extinction coefficient of reduced cytochrome c, at 550 nm, including in 10.1074/jbc.M203118200; 10.2116/analsci.8.829; 10.1016/0076-6879(67)10064-5. A new reference is included in manuscript.
Comment 7: 2.3.4. Super-complex I-III activity. Do you count Complex I-III activity as a super-complex here?
Comment 8: 2.3.5. Complex II-III conjugated activity, why Complex II-III conjugated activity.
Answer to comments 7 and 8: It is reported for Mammalia mitochondria that the transmembrane complexes I, III and IV can occurs as individual identities as well as organized in super-complexes with well-defined proportions. On the other hand, the succinate dehydrogenase, also designed as complex II, is an enzyme of the citric acid cycle that are connected to the matrix bilayer leaflet of inner mitochondrial membrane and supply the mitochondrial redox chain with electrons resulting from succinate oxidation. Thus, Complex II is mainly committed to the citric acid cycle that occurs in the matrix, so it occurs preferentially as individual enzyme exhibiting low tendency to integrate the super-complexes (https://doi.org/10.1007/s10863-008-9167-5; https://doi.org/10.1016/j.bbabio.2009.12.013). Thus, we consider suitable the use of the terminology of super-complex for the assessment of the integrated Complex I-III activity and only conjugated activity for the integrated activity of Complex II-III. This idea is clarified in the new version of manuscript.
Reviewer 2 Report
In the manuscript entitled “An experimental approach to address the functional relationship between antioxidant enzymes and mitochondrial respiratory complexes” authors have developed a method to study the biological activities of anti-oxidant enzymes in mitochondria-free cytosolic fractions and the mitochondrial complexes I, II, IV as well as conjugated activity of complexes I-III and complexes II-III in the mitochondria-rich fraction from the same cell lysates. Authors have meticulously standardized all enzymatic assays with controls and tried to investigate the functional relationship between antioxidant enzymes and mitochondrial complexes.
Comments:
1) To determine the functional relationship between antioxidant enzymes and mitochondrial complexes, it would have been best if authors compared the enzymatic activities of cells treated with stimuli (to cause oxidative stress) and without stimuli. Because in most of the disease state there is uprise of ROS that affects the antioxidant enzyme system and mitochondrial respiratory chain complexes as well. In another case, authors could have selected two different cells with various level of ROS.
2) For the better comparison of antioxidant enzymatic activities, all the enzymes should have same enzymatic unit. Cu/Zn-SOD has U/mg whereas GR, GPx and CAT have nmol/min/mg of protein.
3) If possible, error bar can be included to indicate the reproducibility of the time kinetics of the all the enzymatic activities.
4) Authors have proposed the experimental approach to isolate the mitochondria-free cytoplasmic fractions and mitochondria-rich fractions from a single T-25 or T-75 flasks. It is not clear whether authors assessed the purity of these subcellular fractions using western blot or other methods to avoid cross contamination of mitochondria.
5) Although manuscript is written well, however there are many sentences need to be corrected grammatically and several sentences are too long and difficult to understand.
Author Response
General comment: In the manuscript entitled “An experimental approach to address the functional relationship between antioxidant enzymes and mitochondrial respiratory complexes” authors have developed a method to study the biological activities of antioxidant enzymes in mitochondria-free cytosolic fractions and the mitochondrial complexes I, II, IV as well as conjugated activity of complexes I-III and complexes II-III in the mitochondria-rich fraction from the same cell lysates. Authors have meticulously standardized all enzymatic assays with controls and tried to investigate the functional relationship between antioxidant enzymes and mitochondrial complexes.
Comment 1: To determine the functional relationship between antioxidant enzymes and mitochondrial complexes, it would have been best if authors compared the enzymatic activities of cells treated with stimuli (to cause oxidative stress) and without stimuli. Because in most of the disease state there is uprise of ROS that affects the antioxidant enzyme system and mitochondrial respiratory chain complexes as well. In another case, authors could have selected two different cells with various level of ROS.
Answer to comment 1: Although the reviewer’s comment is an exciting perspective for research papers, the present work only aims to report experimental methodologies to assess the activities of cytosolic antioxidant enzymes and mitochondrial complexes in same cell preparation. A set of assays that provide suitable tools to correlate the performance of the enzymatic antioxidant system in cytosol fraction with activity of mitochondrial respiratory complexes. Tools that can be used to understand the response of cells to oxidant stimuli (or other stress agents) as well as to compare different types of cells under the control conditions. This now clarified in the new version of the manuscript.
Comment 2: For the better comparison of antioxidant enzymatic activities, all the enzymes should have same enzymatic unit. Cu/Zn-SOD has U/mg whereas GR, GPx and CAT have nmol/min/mg of protein.
Answer to comment 2: As suggested, the activity of Cu/Zn-SOD enzyme was changes for nmol/min./mg of protein.
Comment 3: If possible, error bar can be included to indicate the reproducibility of the time kinetics of the all the enzymatic activities.
Answer to comment 3: We recognize the scientific relevance of comment. However, as referred in the manuscript, a typical result for each enzyme is plotted in graph and the cells of 1 flask allow to obtain an amount of mitochondria-rich fraction and mitochondria-free cytosolic fraction to performer the described assays but not to reply all of them, thereby it is not possible to present error bars. However, to show the reproducibility of the methodologies, the activity of each enzyme, expressed as mean ± std, of three independent assays performed with mitochondria-free cytosolic fraction or mitochondria-rich fraction obtained from cells of three consecutive passages were included in Tables 11 and 12.
Comment 4: Authors have proposed the experimental approach to isolate the mitochondria-free cytoplasmic fractions and mitochondria-rich fractions from a single T-25 or T-75 flasks. It is not clear whether authors assessed the purity of these subcellular fractions using western blot or other methods to avoid cross contamination of mitochondria.
As indicated in Figure 2, citrate synthase activity was assessed in supernatant, and mitochondria-free cytosolic fraction in addition to mitochondria-rich fraction to evaluate the purity of the fractionation methodology. In fact, the activity of this enzyme is not detected in mitochondria-free cytosolic fraction and the activity of this mitochondrial enzyme detected in supernatant is preserved in mitochondria-rich fraction.
Answer to comment 4: We can assure that the mitochondria-free cytosolic fraction does not contain mitochondria, since no citrate synthase activity was detected in this fraction. Moreover, the activity of this mitochondrial enzyme detected in supernatant (see Figure 2) is preserved in mitochondria-rich fraction, reason why we chose this designation. This information is included in the new version of the manuscript.
Comment 5: Although manuscript is written well, however there are many sentences need to be corrected grammatically and several sentences are too long and difficult to understand.
Answer to comment 5: The issues present in this comment were taken into account and the manuscript was revised, the modifications are highlighted in track changes in the new version of the manuscript.
Reviewer 3 Report
In this manuscript, Mendes and colleagues describe experimental procedures for simultaneously measuring the activity of mitochondrial respiratory chain enzymes and antioxidant enzymes. A fundamental understanding of how the balance of both activities is maintained is of great importance, since this balance is a prerequisite for health and, conversely, imbalance underlies the development of various pathological conditions.
Overall, the manuscript is well structured and coherently written, and explains nicely the rationale for its preparation as well as the methods used. In terms of reproducibility, the manuscript would benefit from some edits, such as the use of final concentrations instead of (or in addition to) applied volumes and the addition of catalog numbers for major ingredients and/or instrument (example below).
Specific comments/questions:
The authors have used Dulbecco's Modified Eagle Medium/Nutrient Mixture F-12, which according to a manufacturer combines DMEM's high concentrations of glucose, amino acids, and vitamins with the wide variety of components found in Ham’s F12 media. Did the authors consider using low glucose (or galactose) media to study cells that are presumably more oxidative?
Please include the necessary information to facilitate replication elsewhere, e.g. catalog numbers of chemicals or the exact volume and fit ( clearance) of the glass-on-Teflon potter. Handling is also important, e.g., how many strokes were applied at what speed for each round, if this process was facilitated by a drill. To maintain readability, the authors could add catalog numbers and other useful information in a separate table.
Did the authors consider the activity of manganese superoxide dismutase (MnSOD/SOD2) in the mitochondrial fraction? And if not, was it due to lack of material or some other technical issue.
Is it known whether the freeze/thaw cycles used affected the stoichiometry of the respiratory (super)complexes and thus possibly the accessibility of the substrates to the active site? Did the authors consider dodecyl maltoside (DDM) or digitonin for solubilization to preserve or destroy the supercomplexes? If not, would this be a useful approach or is there another technical issue?
Minor issues:
Abstract: “…The present article gathers the experimental procedures to ob- 18 tain, from isolated cells, a mitochondria-free cytosolic fraction and a mitochondria-rich fraction, 19 alongside the methodologies to evaluate the activity of the main antioxidant enzymes, superoxide 20 dismutase, catalase, glutathione reductase and glutathione peroxidase in the cytosolic fraction, and 21 the activity of the individual mitochondrial complexes I, II, IV, as well as, the conjugated activity of 22 complexes I-III and complexes II-III in the mitochondria-rich fraction…”
This sentence is far too long and difficult to read and grasp. I would suggest splitting it into two sentences.
Line 84: “… tested with normal human fibroblast …”.
Please delete “normal”.
Lines 231-233: “The assessment of complex III activity represents a different challenge since its substrate (ubiquinone) is not soluble in water.”
Would complex III be measurable with a less hydrophobic substrate such as Q2?
Lines 290-292: “Thus, the time-dependent decrease of absorbance intensity at 600nm (ԐDCPIP = 20700M-1.cm-1) in the absence and presence of oxaloacetate (Complex II inhibitor) is recorded in a microplate reader (Table 6).”
Is there a technical reason why malonate was not used?
Line 401: “… processed by GPx enzyme under a catalytic cycle that uses the reduced power of GSH …”
Did the authors mean “… reducing power of GSH …”?
Lines 497, 498: “… Complex I is considered the primary source of mitochondrial ROS 497 generation and the process is mainly explained by a one-electron redox event …”
There is good evidence that complex III is also an important source of ROS. Please discuss.
Author Response
General comment: In this manuscript, Mendes and colleagues describe experimental procedures for simultaneously measuring the activity of mitochondrial respiratory chain enzymes and antioxidant enzymes. A fundamental understanding of how the balance of both activities is maintained is of great importance, since this balance is a prerequisite for health and, conversely, imbalance underlies the development of various pathological conditions. Overall, the manuscript is well structured and coherently written, and explains nicely the rationale for its preparation as well as the methods used. In terms of reproducibility, the manuscript would benefit from some edits, such as the use of final concentrations instead of (or in addition to) applied volumes and the addition of catalog numbers for major ingredients and/or instrument (example below).
Specific comments/questions:
Comment 1: The authors have used Dulbecco's Modified Eagle Medium/Nutrient Mixture F-12, which according to a manufacturer combines DMEM's high concentrations of glucose, amino acids, and vitamins with the wide variety of components found in Ham’s F12 media. Did the authors consider using low glucose (or galactose) media to study cells that are presumably more oxidative?
Answer to comment 1: Although the reviewer’s comment has high relevance for research papers, the present work only aims to report experimental methodologies to assess the activities of cytosolic antioxidant enzymes and mitochondrial complexes in same cell preparation. Thus, it can be made with cells growth in any suitable medium. Thus, the effects of the above-mentioned cell growth conditions/stimuli were not assessed in the present paper. However, in a previous study we have reported how Caco-2 cells adapt their metabolic activity to aglycemic nutritional condition (glucose was replaced by galactose) considering bioenergetics and lipidomic approaches. (https://doi.org/10.1007/s10863-013-9531-y).
Comment 2: Please include the necessary information to facilitate replication elsewhere, e.g. catalog numbers of chemicals or the exact volume and fit (clearance) of the glass-on-Teflon potter. Handling is also important, e.g., how many strokes were applied at what speed for each round, if this process was facilitated by a drill. To maintain readability, the authors could add catalog numbers and other useful information in a separate table.
Answer to comment 2: In fact, replication of results is essential in science. Regarding the chemical compounds used, the important is to know their purity degree, all the compounds used have analytic degree (purity ≥ 99%) and were purchased in SIGMA/MERCK group. This information is included in the new version of the manuscript. Regarding the Potter-Elvehjem homogenizer issue, also addressed by Reviewer 1, we state that in the present work, cell suspensions were homogenized in a 3 mL Potter-Elvehjem homogenizer (Merck, Reference P7734-1EA; clearance of 0.004 - 0.006 in.) using 20 up- and downstrokes at 500 rpm.
Comment 3: Did the authors consider the activity of manganese superoxide dismutase (MnSOD/SOD2) in the mitochondrial fraction? And if not, was it due to lack of material or some other technical issue.
Answer to comment 3: We agree with the relevance of the reviewer's comment, since Mn-SOD is an isoform of SOD enzyme that only occurs in the mitochondria. The assays described in manuscript also allow to assess the activity of Mn-SOD in mitochondria-rich fraction, using a reaction medium supplemented with a specific inhibitor of Cu/Zn-SOD. However, in general the amount of mitochondria-rich fraction obtained with the cells of one flask is fully used in the assessment of the activity of mitochondrial complexes described in the manuscript. Thus, the paper was rationalized to correlate the activity of mitochondrial complexes with antioxidant enzymes activity in mitochondria-free cytosolic fraction.
Comment 4: Is it known whether the freeze/thaw cycles used affected the stoichiometry of the respiratory (super)complexes and thus possibly the accessibility of the substrates to the active site? Did the authors consider dodecyl maltoside (DDM) or digitonin for solubilization to preserve or destroy the supercomplexes? If not, would this be a useful approach or is there another technical issue?
Answer to comment 4: As indicated by Reviewer’ comment, the freeze/thaw cycles in addition to promote inner mitochondrial membrane permeabilization to polar enzyme substrates (e.g., NADH) it can also promote a membrane rearrangement with impact on the stoichiometry of the respiratory (super)complexes. However, as far as we know, the bilayer structure is preserved, which cannot be ensured with the addition of detergents. Thus, the use of dodecyl maltoside (DDM) require for each biological sample a progressive titration to find the minor amount necessary to permeabilize the membrane without promotes changes in the architecture of the mitochondria. Considering the amount of biological sample available, this type of procedure may compromise the feasibility of assessing the activity of the mitochondrial respiratory complexes. Digitonin is not suitable to permeabilize the inner membrane of mitochondria since this membrane does not has cholesterol.
Minor issues:
Abstract: “…The present article gathers the experimental procedures to ob- 18 tain, from isolated cells, a mitochondria-free cytosolic fraction and a mitochondria-rich fraction, 19 alongside the methodologies to evaluate the activity of the main antioxidant enzymes, superoxide 20 dismutase, catalase, glutathione reductase and glutathione peroxidase in the cytosolic fraction, and 21 the activity of the individual mitochondrial complexes I, II, IV, as well as, the conjugated activity of 22 complexes I-III and complexes II-III in the mitochondria-rich fraction…”
This sentence is far too long and difficult to read and grasp. I would suggest splitting it into two sentences.
Line 84: “… tested with normal human fibroblast …”.
Please delete “normal”.
Answer: As recommended, these minor issues were considered in the new version of the manuscript.
Lines 231-233: “The assessment of complex III activity represents a different challenge since its substrate (ubiquinone) is not soluble in water.”
Would complex III be measurable with a less hydrophobic substrate such as Q2?
Answer: Although the subject is relevant, we never approached this question experimentally and we did not find the use of Q2 as a complex III substrate.
Lines 290-292: “Thus, the time-dependent decrease of absorbance intensity at 600nm (ԐDCPIP = 20700M-1.cm-1) in the absence and presence of oxaloacetate (Complex II inhibitor) is recorded in a microplate reader (Table 6).”
Is there a technical reason why malonate was not used?
Answer: This issue was also addressed by reviewer 1. The oxaloacetate, a competitive inhibitor of the Complex II with a strong affinity for succinate-binding site of the enzyme as indicated by the constant inhibition of 1.5 μM (10.1104/pp.70.6.1577), is a suitable for assay proposal. We can also assert that the use of malonate (another competitive inhibitor) is also suitable for assay.
Line 401: “… processed by GPx enzyme under a catalytic cycle that uses the reduced power of GSH …”
Did the authors mean “… reducing power of GSH …”?
Answer: The suggestion was considered in the new version of the manuscript.
Lines 497, 498: “… Complex I is considered the primary source of mitochondrial ROS 497 generation and the process is mainly explained by a one-electron redox event …”
There is good evidence that complex III is also an important source of ROS. Please discuss.
Answer: It is an important issue, but its discussion is beyond the scope of this article. Thus, it was only included a sentence with a reference to associate complex III with mitochondrial ROS production (10.1016/j.bbabio.2012.12.002).
Round 2
Reviewer 1 Report
Dear authors, thank you for your comprehensive answers. Good job.
Reviewer 2 Report
Authors have satisfactorily responded all the questions.